# Improved Analysis for Bandit Learning in Matching Markets

**Fang Kong**
Southern University of Science and Technology
kongf@sustech.edu.cn

**Zilong Wang**
Shanghai Jiao Tong University
wangzilong@sjtu.edu.cn

**Shuai Li**[*]
Shanghai Jiao Tong University
shuaili8@sjtu.edu.cn

## Abstract

A rich line of works study the bandit learning problem in two-sided matching markets, where one side of market participants (players) are uncertain about their preferences and hope to find a stable matching during iterative matchings with the other side (arms). The state-of-the-art analysis shows that the player-optimal stable regret is of order $O(K \log T/\Delta^2)$ where $K$ is the number of arms, $T$ is the horizon and $\Delta$ is the players' minimum preference gap. However, this result may be far from the lower bound $\Omega(\max\{N \log T/\Delta^2, K \log T/\Delta\})$ since the number $K$ of arms (workers, publisher slots) may be much larger than that $N$ of players (employers in labor markets, advertisers in online advertising, respectively). In this paper, we propose a new algorithm and show that the regret can be upper bounded by $O(N^2 \log T/\Delta^2 + K \log T/\Delta)$. This result removes the dependence on $K$ in the main order term and improves the state-of-the-art guarantee in common cases where $N$ is much smaller than $K$. Such an advantage is also verified in experiments. In addition, we provide a refined analysis for the existing centralized UCB algorithm and show that, under $\alpha$-condition, it achieves an improved $O(N \log T/\Delta^2 + K \log T/\Delta)$ regret.

## 1 Introduction

The two-sided matching market problem has been extensively studied in the literature due to its wide range of applications like labor market, school admission, house allocation, and online advertising [26, 9, 1]. There are two sides of participants in the market, such as the employers and workers in the labor market, advertisers and publishers in online advertising. Each participant on the one side has a preference ranking over the other side. The concept of stability, which characterizes the equilibrium state of the market where no participant wants to break up the current matching relationship and find another partner, has attracted great interest from researchers [26]. Achieving stability is critical for ensuring the long-term viability of the market.

A rich line of works [9, 15, 25] study how to find a stable matching in the market. Most of them assume the preference ranking of each market participant is known beforehand, which we refer to as the *offline* setting. However, in real applications, the knowledge of the preferences may be uncertain. For example, in the labor market, employers usually do not know the working abilities of workers before being matched, and advertisers also do not know the exact conversion rate of placing the advertisement in a publisher slot. This makes the traditional algorithms unavailable to

---

[*]Corresponding author.

38th Conference on Neural Information Processing Systems (NeurIPS 2024).

find an exact stable matching. With the emergence of online market platforms such as the online labor market UpWork and TaskRabbit as well as online advertising platforms where employers or advertisers have many similar tasks, market participants are able to learn their unknown preferences during iterative matchings with the other side of agents.

Multi-armed bandit (MAB) is a classic framework that characterizes the learning process during iterative interactions [3, 19]. It considers the setting with single player on one side and multiple arms on the other side. At each round, the player selects an arm and receives a reward. The player has unknown preferences over arms and would learn this knowledge based on the collected rewards. To accumulate as many rewards as possible, the player faces the dilemma of exploration and exploitation. The former selects arms with less observations while the latter focuses on arms with better historical performances. How to balance the exploration and exploitation trade-off is the key of bandit algorithm design. The upper confidence bound (UCB) [3], Thompson sampling (TS) [14, 2], and explore-then-commit (ETC) [10] are common strategies in MAB to achieve this objective.

Liu *et al.* [20] introduce the bandit learning problem in matching markets and try to provide theoretical guarantees. Two sides of agents in the market can be modeled as players and arms. Without loss of generality, denote $N$ and $K$ as the number of players and arms, respectively. It is worth noting that this work and all of the following works assume $N \leq K$ to ensure each player has a chance to be matched. In this problem, the objective is to find a stable matching and minimize the stable regret for each player, which is defined as the difference between the reward of the stable arm and that the player receives during the horizon. Since there may be more than one stable matching, they mainly focus on the players' most preferred one corresponding to the player-optimal stable matching and the least preferred one corresponding to the player-pessimal stable matching. Note that players receive more rewards in the player-optimal stable matching and thus the former objective is the most desirable. Liu *et al.* [20] first study a centralized setting where a central platform would compute allocations for players to avoid conflicts. Both ETC and UCB-type algorithms are proposed for this setting. The former achieves a player-optimal stable regret guarantee with prior knowledge of players' minimum preference gap $\Delta$ and the latter can only ensure to reach the player-pessimal stable matching. Motivated by real applications where the central platform may not always exist, a rich line of works then study the decentralized case where no platform coordinates players' behavior [21, 28, 4, 18, 22]. This line of works again only achieve guarantees for player-pessimal stable regret [21, 18, 28, 4, 22]. Table 1 compares settings and regrets among these works. Until recently, Zhang *et al.* [31] and Kong and Li [16] independently derive algorithms that have polynomial player-optimal stable regret and show the upper bound is $O(K \log T/\Delta^2)$. However, this result may be still far from the lower bound $\Omega(\max\{N \log T/\Delta^2, K \log T/\Delta\})$ [28] since $K$ is usually much larger than $N$ such as that the number of workers (publisher slots) is usually much larger than that of employers in labor markets (advertisers in online advertising, respectively).

In this paper, we try to provide more efficient algorithms and improve the results over existing works. The detailed contribution can be summarized as follows: (1) We propose an algorithm named adaptively explore-then-Gale-Shapley (AETGS) with elimination. State-of-the-art works [31, 16] explicitly separate the exploration and exploitation processes, which can lead to unnecessary regret, as exploring certain preference rankings may not contribute to the exploitation process. To avoid excessive exploration, our AETGS with elimination algorithm integrates the players' learning process into the GS steps. Players adaptively switch between exploration and exploitation and promptly eliminate sub-optimal arms. (2) We prove that the player-optimal stable regret of AETGS with elimination can be upper bounded by $O(N^2 \log T/\Delta^2 + K \log T/\Delta)$. This is the first result that removes the dependence on $K$ in the main regret order term and improves existing works in common cases where $N$ is much smaller than $K$. We also conduct experiments to show the advantages of the algorithm. (3) We refine the analysis of the centralized UCB algorithm in Liu *et al.* [20] for markets satisfying the $\alpha$-condition. By investigating the preference hierarchy structure of the $\alpha$-condition, we demonstrate that the stable matching converges sequentially from player 1 to player $N$. Through inductive analysis over players, we establish an $O(N \log T/\Delta^2 + K \log T/\Delta)$ regret upper bound, which improves the original result for this algorithm in this specific market.

## 2   Related Work

The problem of bandit learning in matching markets is first introduced by Das and Kamenica [8]. They study the special case where both sides of agents have the same preferences and propose

| | Regret bound | Setting |
|---|---|---|
| Liu *et al.* [20] | $O\left(K\log T/\Delta^2\right)*\#$ 
 $O\left(NK\log T/\Delta^2\right)\#$ | known $\Delta$, $\mathrm{gap}_1$ 
 $\mathrm{gap}_2$ |
| Liu *et al.* [21] | $O\left(\dfrac{N^5 K^2 \log^2 T}{\varepsilon^{N^4}\Delta^2}\right)$ | $\mathrm{gap}_2$ |
| Sankararaman *et al.* [28] | $O\left(NK\log T/\Delta^2\right)$ 
 $\Omega\left(\max\left\{N\log T/\Delta^2, K\log T/\Delta\right\}\right)$ | serial dictatorship, $\mathrm{gap}_1$ |
| Basu *et al.* [4] | $O\left(K\log^{1+\varepsilon}T + 2^{\left(\frac{1}{\Delta^2}\right)^{\frac{1}{\varepsilon}}}\right)*$ 

 $O\left(NK\log T/\Delta^2\right)$ | $\mathrm{gap}_2$ 

 $\alpha$-condition, $\mathrm{gap}_1$ |
| Maheshwari *et al.* [22] | $O\left(CNK\log T/\Delta^2\right)$ | $\alpha$-reducible condition, $\mathrm{gap}_1$ |
| Kong *et al.* [18] | $O\left(\dfrac{N^5 K^2 \log^2 T}{\varepsilon^{N^4}\Delta^2}\right)$ | $\mathrm{gap}_2$ |
| Zhang *et al.* [31] | $O\left(K\log T/\Delta^2\right)*$ | $\mathrm{gap}_2$ |
| Kong and Li [16] | $O\left(K\log T/\Delta^2\right)*$ | $\mathrm{gap}_3$ |
| Ours | $O\left(N^2\log T/\Delta^2 + K\log T/\Delta\right)*$ 
 $O\left(N\log T/\Delta^2 + K\log T/\Delta\right)\#$ | $\mathrm{gap}_4$ 
 $\alpha$-condition, $\mathrm{gap}_3$ |

Table 1: Comparisons of settings and regret bounds with most related works, $*$ represents the player-optimal stable regret and bounds without labeling $*$ are for player-pessimal stable regret, $\#$ represents the centralized setting. $N$ and $K$ are the number of players and arms with $N \leq K$, $T$ is the total horizon, $\Delta$ corresponds to some preference gap, $\varepsilon$ depends on the hyper-parameter of algorithms, and $C$ is related to the unique stable matching condition which can grow exponentially in $N$. The definition of $\Delta$ in different works requires particular care. We use $\mathrm{gap}_1, \mathrm{gap}_2, \mathrm{gap}_3, \mathrm{gap}_4$ represent the minimum preference gap between the (player-optimal) stable arm and the next arm after the stable arm in the preference ranking among all players, the minimum preference gap between any different arms among all players, the minimum preference gap between the first $N+1$ ranked arms among all players, and the minimum preference gap between arms that are more preferred than the next of the player-optimal stable arm among all players, respectively. Based on the property that the player-optimal stable arm of each player must be its first $N$-ranked (shown in Appendix), there would be $\mathrm{gap}_1 \geq \mathrm{gap}_4 \geq \mathrm{gap}_3 \geq \mathrm{gap}_2$. So our dependence on $\Delta$ is better than the state-of-the-art works [31, 16] for general markets.

some empirical methods to solve the problem. Liu *et al.* [20] first theoretically formulate this problem and provide an upper bound for the stable regret of players. They propose a centralized explore-then-commit (ETC) algorithm and upper confidence bound (UCB) algorithm, which obtain an $O\left(K\log T/\Delta^2\right)$ player-optimal stable regret and $O\left(NK\log T/\Delta^2\right)$ player-pessimal stable regret, respectively. It is worth noting that the former ETC algorithm requires knowledge about $\Delta$ to ensure the algorithmic operation. Due to the generality, the following works focus on the decentralized setting. Liu *et al.* [21] and Kong *et al.* [18] propose the UCB and TS-type algorithm for general decentralized markets, respectively. Such a setting is much more challenging and both of them only achieve $O\left(\exp(N^4)N^5 K^2 \log^2(T)/\Delta^2\right)$ upper bound for the player-pessimal stable regret.

To improve the stable regret guarantee, a line of research studies some special markets with unique stable matching in which case the player-optimal stable matching is equivalent to the player-pessimal one. Sankararaman *et al.* [28] propose the UCB-D3 algorithm based on the assumption of serial dictatorship, i.e., all arms share the same preferences, and obtain an $O\left(NK\log T/\Delta^2\right)$ regret upper bound. To investigate the problem hardness, they also derive a lower bound $\Omega\left(\max\left\{N\log T/\Delta^2, K\log T/\Delta\right\}\right)$ under this assumption. Basu *et al.* [4] consider more general $\alpha$-condition setting for unique stable matching. They propose the UCB-D4 algorithm and also achieve the $O\left(NK\log T/\Delta^2\right)$ regret bound. Later, Maheshwari *et al.* [22] study the market satisfy-

ing $\alpha$-reducible condition and proposes a communication-free algorithm. Their regret bound has an exponential dependence on the number of market participants. Recently, researchers have developed algorithms that can achieve player-optimal stable regret guarantees without assuming unique stable matching. Both Zhang *et al.* [31] and Kong and Li [16] propose ETC-type algorithms that achieve $O\left(K \log T / \Delta^2\right)$ player-optimal stable regret. Wang and Li [29] studies the matching markets with serial dictatorship and obtains the $O\left(N \log T / \Delta^2 + K \log T / \Delta\right)$ regret. Table 1, compares our proposed algorithm with these related works in terms of their corresponding settings and theoretical guarantees.

There are also other works considering unknown preferences in matching markets. Wang *et al.* [30] study a many-to-one market where an arm can accept multiple players. Jagadeesan *et al.* [12] consider online matching markets with monetary transfers. Min *et al.* [23] investigate Markov matching markets where state transitions occur during the matching process and players' rewards depend on the current state. Other studies have focused on non-stationary rewards, such as Muthirayan *et al.* [24], Ghosh *et al.* [11], who propose robust algorithms to mitigate the impact of reward disturbances. Additionally, several studies have explored the problem of offline matching market learning. Dai and Jordan [6, 7] propose approaches that leverage historical data to design optimal matching or recommend participants on both sides.

## 3 Setting

This paper considers the problem of bandit learning in two-sided matching markets. Denote $\mathcal{N} = \{p_1, p_2, \ldots, p_N\}$ as the player set and $\mathcal{K} = \{a_1, a_2, \ldots, a_K\}$ as the arm set. Let $N$ and $K$ be the number of players and arms, respectively. To ensure that each player can be matched with an arm, we follow previous works and assume $N \leq K$ [20, 21, 28, 4, 18, 31, 16].

For each player $p_i \in \mathcal{N}$, its preference towards arm $a_j$ can be portrayed by an absolute utility $\mu_{i,j} \in (0, 1]$. For any pair of arms $a_j$ and $a_{j'}$, $\mu_{i,j} > \mu_{i,j'}$ indicates that player $p_i$ prefers arm $a_j$ over $a_{j'}$. Following previous works for matching markets [9, 20, 21, 28, 4, 18, 31, 16], players are assumed to have distinct preferences over different arms, i.e., $\mu_{i,j} \neq \mu_{i,j'}$ for any $a_j \neq a_{j'}$. In practice, players' preferences which correspond to workers' abilities and the publisher's conversion rates are typically unknown and can be learned through the interactive matching process. On the other side, each arm $a_j$ also has a fixed and distinct preference utility $\pi_{j,i}$ over each player $p_i \in \mathcal{N}$, and $\pi_{j,i} > \pi_{j,i'}$ means that arm $a_j$ prefers player $p_i$ over $p_{i'}$. As in labor markets where workers usually know their preferences over employers based on the payments and task types, the preferences of arms are assumed to be known beforehand [20, 21, 18, 28, 4, 31, 16].

At each round $t = 1, 2, \ldots$, each player $p_i$ proposes to an arm $A_i(t)$. For each arm $a_j$, denote $A_j^{-1}(t) = \{p_i : A_i(t) = a_j\}$ as the set of players who selects arm $a_j$ at round $t$. When more than one player selects $a_j$, it accepts its most-preferred one in $A_j^{-1}(t)$, i.e. $a_j$ will match with $p_i \in \arg\max_{p_i \in A_j^{-1}(t)} \pi_{j,i}$. If a player $p_i$ is successfully matched with arm $A_i(t)$, it will receive a random reward $X_i(t)$ characterizing its matching experience, which we assume is a 1-subgaussian random variable with expectation $\mu_{i,A_i(t)}$. Otherwise, $p_i$ is rejected by its proposed arm and only gets reward $X_i(t) = 0$. Denote $\bar{A}_i(t)$ as the final matched arm of player $p_i$ at round $t$. Then $\bar{A}_i(t) = A_i(t)$ if $p_i$ is accepted by the arm $A_i(t)$ and we simply set $\bar{A}_i(t) = \emptyset$ if $p_i$ is rejected.

Stability is a key property of a matching in two-sided markets [9, 27, 25]. A matching $\bar{A}(t) = \{(i, \bar{A}_i(t)) : i \in [N]\}$ is stable if no market participant wants to break up its current matching relationship and find a new partner. Formally speaking, there is no player-arm pair $(p_i, a_j)$ such that $\mu_{i,j} > \mu_{i,\bar{A}_i(t)}$ and $\pi_{j,i} > \pi_{j,\bar{A}_j^{-1}(t)}$. It is worth noting that there may be multiple stable matchings in the market. Denoted $M = \{m : m \text{ is stable}\}$ as the set of all stable matchings. It is shown that there exists a stable matching $m^* \in M$ such that all players are matched with their most preferred stable arm [9], i.e., $\mu_{i,m_i^*} \geq \mu_{i,m_i}$ for any $m \in M, i \in [N]$. Given a specified horizon $T$, the learning objective is to minimize the player-optimal stable regret for each player $p_i$ which is defined as the difference between the cumulative reward received by being matched with $m_i^*$ and the cumulative reward received by $p_i$ over $T$ rounds:

$$Reg_i(T) = \mathbb{E}\left[\sum_{t=1}^{T} \left(\mu_{i,m_i^*} - X_i(t)\right)\right].$$

Here, the expectation is taken over by the randomness of the reward generation and the randomness inherent in the player's strategy.

For completeness, we introduce the procedure of the offline Gale-Shapley (GS) algorithm, which would be useful when describing the algorithmic details. The offline GS algorithm is a classic algorithm to find the player-optimal stable matching when both sides of the market participants know their exact preference rankings. Following offline GS, each player proposes to the arm one by one based on its preference ranking. Until no rejection happens, the final matching is exactly the player-optimal stable matching [9]. Specifically, at the first step, all players propose to their most preferred arm. Arms would accept their most preferred player among those who propose to it and reject others. Then players who are rejected at previous steps would then propose to their next preferred arm. And arms still reject the players who propose to it except for their most preferred one. Such a process continues until no rejection happens.

For convenience, we also define some useful notations that quantify the hardness of the learning problem in matching markets and are used in the later analysis.

**Definition 3.1.** For each player $p_i$, denote $\sigma_i$ as $p_i$'s preference ranking and let $\sigma_{i,k}$ as $p_i$'s the $k$-th preferred arm in its ranking. With a little abuse of notation, let $\sigma_i(a_j)$ represent the rank of arm $a_j$ in $p_i$'s preference. For each player $p_i$ and arm $a_j \neq a_{j'}$, let $\Delta_{i,j,j'} = |\mu_{i,j} - \mu_{i,j'}|$ be the preference gap of $p_i$ between $a_j$ and $a_{j'}$. Define $\Delta = \min_{i,k\in[\sigma_i(m_i^*)]} \Delta_{i,\sigma_{i,k},\sigma_{i,k+1}}$ as the minimum preference gap between the arm ranked the first $(\sigma_i(m_i^*) + 1)$-th among all players. Further, define $\Delta_N = \min_{i,k\in[N]} \Delta_{i,\sigma_{i,k},\sigma_{i,k+1}}$ as the minimum preference gap between the arm ranked the first $(N + 1)$-th among all players.

## 4  Algorithm for General Markets

In this section, we propose an algorithm called adaptively explore-then-Gale-Shapley (AETGS) with elimination. For simplicity, we present the centralized version of the algorithm in Algorithm 1 from view of player $p_i$. The discussion on how to extend it to a decentralized version is deferred to later subsections.

In general, AETGS with elimination is an adaptive version of the GS algorithm. Since players do not know their preference rankings, they need to learn this knowledge by exploring arms (Line 4). To reduce the regret during exploration, players would adaptively eliminate sub-optimal arms (Line 6-8). Until they find their most preferred arm among available arms, they will stop exploration and focus on this arm (Line 9-11). And once the player finds this arm is occupied by a more preferred player, it would re-start exploration to find the next preferred arm (Line 12-19).

Specifically, each player $p_i$ still maintains $\hat{\mu}_{i,j}(t)$ and $T_{i,j}(t)$ to represent the empirical mean and the number of observations on each arm $a_j$ at the end of round $t$. To determine whether an arm is more preferred than another, it maintains a confidence interval for each arm $a_j$ with upper bound $\text{UCB}_{i,j}(t) := \hat{\mu}_{i,j}(t) + \sqrt{6\log T/T_{i,j}(t)}$ and lower bound $\text{LCB}_{i,j}(t) := \hat{\mu}_{i,j} - \sqrt{6\log T/T_{i,j}(t)}$. When $T_{i,j} = 0$, they will be initialized as $+\infty$ and $-\infty$, respectively. And once the confidence intervals of the two arms are disjoint, it can regard the arm with a higher empirical mean to be more preferred (Line 1). To be consistent with the offline GS, each player maintains $\mathcal{D}_i$ to represent the set of arms that have rejected $p_i$ during previous steps. In the beginning, it is initialized as an empty set. And we use $\mathcal{A}_i$ to represent the available arms with the potential to be the stable arm of $p_i$, which is initialized as $\mathcal{K} \setminus \mathcal{D}_i$. For convenience, denote $\text{E}_i$ as the exploration status of player $p_i$. $\text{E}_i = \text{True}$ means that $p_i$ still needs to explore arms in $\mathcal{A}_i$ to determine its most preferred arm. And $\text{E}_i = \text{False}$ means that $p_i$ already finds its most preferred arm and now focuses on this arm (Line 2).

To reduce the regret suffered during exploration, players would update $\mathcal{A}_i$ and eliminate sub-optimal arms in real-time (Line 6-8). Here to avoid collision during round-robin exploration, we would maintain $\mathcal{A}_i$ such that it contains no less than $N$ arms if $p_i$ still has not determined its most preferred one. Thus the union of the available arm set over all players with $\text{E}_i = \text{True}$ contains more than $N$ arms and the round-robin exploration over $\mathcal{A}_i$ for each such player $p_i$ can be carried out without collisions. For completeness, we defer how to arrange players' explorations in later discussions. And once there exists an arm in $\mathcal{A}_i$ that can be regarded to be optimal, $p_i$ will set the exploration status $\text{E}_i$ to be False and update the exploration arm set $\mathcal{A}_i$ to only contain this optimal arm. For convenience, denote $A_i$ as this arm (Line 9-11).

**Algorithm 1** adaptively explore-then-Gale-Shapley with elimination (from the view of player $p_i$)

**Input:** $N, K, T$.
 1: Initialize: $\hat{\mu}_{i,j}(0) = 0, T_{i,j}(0) = 0, \text{UCB}_{i,j}(0) = \infty, \text{LCB}_{i,j}(0) = -\infty, \forall j \in [K]$;
 2: Initialize: $\mathcal{D}_i = \emptyset, \mathcal{A}_i = \mathcal{K}, \text{E}_i = \text{True}$;
 3: **for** round $t = 1, 2, \cdots, T$ **do**
 4:     Select $A_i(t) \in \mathcal{A}_i$ in a round-robin manner;
 5:     Update $\hat{\mu}_{i,j}(t), T_{i,j}(t)$ as Algorithm 2; compute $\text{UCB}_{i,j}(t), \text{LCB}_{i,j}(t)$ for any $j \in [K]$;
 6:     **if** $|\mathcal{A}_i| > N$ and $\exists j \in \mathcal{A}_i$ s.t. $\text{UCB}_{i,j}(t) < \max_{j' \in \mathcal{A}_i} \text{LCB}_{i,j'}(t)$ **then**
 7:         $\mathcal{A}_i = \mathcal{A}_i \backslash \{j\}$;
 8:     **end if**
 9:     **if** $\exists j \in \mathcal{A}_i$ s.t. $\text{LCB}_{i,j}(t) > \text{UCB}_{i,j'}(t)$ for any $j' \in \mathcal{A}_i$ and $j' \neq j$ **then**
10:         $\mathcal{A}_i = \{j\}, \text{E}_i = \text{False}, A_i = j$;
11:     **end if**
12:     **for** other player $p_{i'}$ with $\text{E}_{i'} = \text{False}, A_{i'} \in \mathcal{A}_i$ **do**
13:         **if** $\pi_{A_{i'},i'} > \pi_{A_{i'},i}$ **then**
14:             $\mathcal{D}_i = \mathcal{D}_i \cup \{A_{i'}\}, \mathcal{A}_i = \mathcal{K} \backslash \mathcal{D}_i$;
15:             **if** $\text{E}_i = \text{False}$ and $A_i \in \mathcal{D}_i$ **then**
16:                 $\text{E}_i = \text{True}$;
17:             **end if**
18:         **end if**
19:     **end for**
20: **end for**

The update of the available arm set should not only depend on $p_i$'s own observations but also on the other market participants. Specifically, if a player $p_{i'}$ determines $A_{i'}$ as its most preferred arm, then the final stable player of $A_{i'}$ would be the same as or more preferred than $p_{i'}$. So if $A_{i'}$ prefers $p_{i'}$ than $p_i$, then $A_{i'}$ would not be the stable arm of $p_i$ and there is no need for $p_i$ to explore $A_{i'}$ anymore. In this case, $p_i$ deletes arm $A_{i'}$ from its available set and update $\mathcal{A}_i$ (Line 12-19). It is worth noting that this operation may incorporate the eliminated arms again in $\mathcal{A}_i$. This is reasonable as the previously eliminated arm may be more preferred than the current arms in $\mathcal{A}_i$ after the deletion operation. And if the deleted arm is $p_i$'s current most preferred arm, it will mark $\text{E}_i$ as True and restart exploration to find the next most preferred one (Line 15-17).

### 4.1 Theoretical Results.

**Theorem 4.1.** *Following Algorithm 1, the player-optimal stable regret for each player $p_i$ satisfies*

$$Reg_i(T) \leq O\left(N^2 \log T / \Delta^2 + K \log T / \Delta\right).$$

Due to the space limit, the proof of Theorem 4.1 is deferred to Appendix A. The following are discussions on the detailed implementation as well as the novelty of the result.

**Arrangement of the round-robin exploration process.** Recall that the number of available arms of each player is always larger than $N$ based on Line 6 and we assume players can explore their available arms in a round-robin manner without conflict. We now propose an arrangement by letting players explore the available arms in units of $N$ to guarantee this property. Specifically, in every $2N$ rounds, each player selects the $N$ available arms with the fewest observations (randomly breaks ties) and explores them in a round-robin way. It can be shown by contradiction that there exists an assignment such that each player can successfully match with their respective $N$ arms once during these $2N$ rounds (Lemma B.2 in Appendix), which only doubles the original regret without influencing the regret order. This guarantees that after every $2N$ rounds, the observation count difference among all available arms is at most 1. Players would perform arm elimination and optimal arm identification (Line 6 and 9) in the end of each $2N$ rounds. Therefore, compared to the timely eliminating/deleting of arms, this approach ensures that each player will select each arm at most one additional time during each exploration cycle before the player finds the optimal one. Since each player may restart exploration (Line 12-13) up to $N^2$ times (each of $N$ players can focus on $N$ arms), this scenario leads to an additional $O(N^2K)$ constant regret and does not influence the final regret order.

**Extension to the decentralized setting.** For simplicity, we present the algorithm in a centralized manner. It can also be extended to the decentralized version where no central platform coordinates players' selections. Specifically, we divide the total horizon into several phases and the length of each grows exponentially, i.e., the lengths of phases are $2, 4, 8, \cdots$. At the end round of each phase, players would decide whether to eliminate sub-optimal arms as Line 6-8, whether to determine one arm as the most preferred one and update the exploration status as Line 9-11. After the end of each phase, players would communicate their current exploration status, update their deletion set and available arm set, re-update the exploration status as Line 12-19, and then communicate their updated available arm set to determine the round-robin exploration process in the next phase. If there exists a player whose exploration status becomes True from False during communication, the phase length would re-start from 2 and grows exponentially since new arms may be required for exploration. If the final exploration status of all players is False and their optimal arms are different, the next phase would continues until the end of the interaction. The detailed implementation of communication is deferred to the next paragraph. Based on the communication, players with $E_i =$ True can have a pre-agreed protocol to explore arms in their available arm set in a round-robin manner without collision as discussed in the last paragraph. Within each phase, they just round-robin explore arms and collect observations but do not make any decision on arms' optimality. If $L$ observations on a sub-optimal arm are enough to decide its sub-optimality in the centralized version, then this arm would be eliminated at the end of the corresponding phase with the selected time to be at most $2L$ due to the exponentially increasing phase length. So the regret in this decentralized version is at most two times as that suffered in the centralized version.

This paragraph describes the implementation of the communication procedure. Recall that players need to communicate their exploration status and available arm set (calculated by subtracting the deletion and eliminating set from $\mathcal{K}$) at the end of each phase. For the phase length, recall that it grows exponentially until a player's exploration status becomes True from False and a player updates $E_i$ from False as True only when its most preferred arm is occupied by a higher-priority player (Line 15). As shown by Lemma A.3, each player may occupy $N$ arms, so such event happens at most $N^2$ times. And when all players find their unique optimal arm which requires $O(N^2 \log T/\Delta^2)$ times, the phase would continue until the end of the interaction. Above all, the total number of phases is of order $O\left(N^2 \log\left(N^2 \log T/\Delta^2\right)\right)$. For the detailed communication procedure, as phase 1 in Kong and Li [16], players can first estimate their unique indices and we assume the matching results are public as [16, 18, 21]. During the communication block of each phase, players sequentially transmit their data based on their indices, received by others through matching outcomes. Specifically, in the corresponding round, player $p_i$ selects the focused arm if $E_i =$ False and nothing otherwise, incurring an $O\left(N^3 \log\left(N^2 \log T/\Delta^2\right)\right)$ cost for status communication in all phases. For deletion (eliminating) sets, $p_i$ first selects the arm with index $k$ to indicate it will transmit $k$ arms and then sequentially selects these $k$ arms. The communication cost on the arm set size is $O\left(N^3 \log\left(N^2 \log T/\Delta^2\right)\right)$. Recall that players delete arms only when a higher-priority player focuses on this arm, so $N$ players focus on at most $N$ arms before reaching stability and each player deletes up to $N$ arms. Also, each player can eliminate up to $K-N$ arms during each exploration and would re-start exploration for at most $N^2$ times. Thus the communication cost on the deletion (eliminating) arms is $O(N^3 K)$ and the total communication cost is $O\left(N^3 \log\left(N^2 \log T/\Delta^2\right) + N^3 K\right)$, which is not the main order of the regret.

**Key idea of removing the dependence on $K$.** Balancing the exploration-exploitation trade-off is the key to achieving low regret. Previous efforts were devoted to addressing pessimal stable regret [20, 21, 18] and uniqueness assumptions [28, 4] using classic UCB and TS strategies. Until recently, Zhang *et al.* [31] and Kong and Li [16] show that ETC-type strategies better fit this problem. Specifically, players first uniformly explore arms to learn the complete preference ranking of the top $N$ arms, and then use the GS procedure for exploitation to find the player-optimal stable matching. However, such a method may over-explore and cause unnecessary regret. The reason is that to learn the first $N$-ranked arms, each sub-optimal arm $a_j$ must be selected $O(\log T/\Delta_{i,\sigma_{i,N},j}^2)$ times to be distinguished from the $N$-ranked arm. And each time selecting this arm, the player pays $\Delta_{i,m_i^*,j}$ regret. The mismatch between the paid regret and the difference to be figured out results in $O(K \log T/\Delta_N^2)$ regret.

In contrast to the existing approach, we present a more adaptive perspective that integrates the learning process into each GS step. To avoid additional regret, players do not need to estimate their complete preference. Instead, they would start exploitation once the optimal available arm is identi-

fied. And if this arm is occupied by a higher-priority player, this player would restart exploration to find the next preferred one. To avoid the additional cost while exploring the optimal arm, we design a more efficient way to let players promptly eliminate $K - N$ sub-optimal arms and only maintain the remaining $N$ arms to guarantee no collision. For the eliminated arm $a_j$, the reward difference to be figured out is at least $\Delta_{i,m_i^*,j}$, which matches the regret when selecting this arm. So the regret caused by these eliminated arms is $O(K \log T / \Delta)$, avoiding dependence on $K$ in the main order.

Recall Kong and Li [17] propose an adaptively-explore-then-deferred-acceptance (AETDA) algorithm for more general many-to-one markets with responsiveness. Compared with their realization in the one-to-one setting, our algorithm shares the same idea of balancing exploration and exploitation but further introduces the elimination operation to avoid unnecessary selections. Compared with their $O(N^2 K \log T / \Delta^2)$ result in the reduced one-to-one setting, our Theorem 4.1 removes the dependence on $K$ in the main order.

**Discussion on the definition of gaps.** Recall that our $\Delta$ is defined as the minimum preference difference among arms that ranked in the first $(\sigma_i(m_i^*) + 1)$-th positions ($\text{gap}_4$), while the lower bound in Sankararaman *et al.* [28] for markets with serial dictatorship depends on $\Delta$ that is defined as the minimum preference difference between the arm ranked $\sigma_i(m_i^*)$ and the arm ranked $\sigma_i(m_i^*) + 1$ ($\text{gap}_1$ in Table 1, respectively). It is an open problem whether the lower bound should depend on our $\text{gap}_4$ in general markets. Here we would like to discuss that the knowledge of $\text{gap}_4$ is important to learn the true stable matching. Consider a market with 4 players and 5 arms. The preference rankings of players are $p_1 : a_1 > a_2 > a_3 > a_4 > a_5; p_2 : a_2 > a_3 > a_1 > a_4 > a_5; p_3 : a_3 > a_1 > a_2 > a_4 > a_5; p_4 : a_1 > a_2 > a_3 > a_4 > a_5$ and the preference rankings of arms are $a_1 : p_2 > p_3 > p_4 > p_1; a_2 : p_3 > p_4 > p_1 > p_2; a_3 : p_4 > p_1 > p_2 > p_3; a_4 : p_1 > p_2 > p_3 > p_4; a_5 : p_1 > p_2 > p_3 > p_4$. In this market, the player-optimal stable matching is $\{(p_1, a_4), (p_2, a_1), (p_3, a_2), (p_4, a_3)\}$. However, if player $p_1$ has collected enough observations to identify $\text{gap}_1$ but not collected enough observations to identify $\text{gap}_4$ and wrongly estimate the first $\sigma_i(m_i^*)$ ranked arms, i.e., $p_1$ wrongly estimate the preference ranking as $p_1 : a_1 > a_2 > a_4 > a_3 > a_5$. Then the computed player-optimal stable matching under this preference ranking is $\{(p_1, a_4), (p_2, a_3), (p_3, a_1), (p_4, a_2)\}$, which is not stable in the original market as player $p_1$ and arm $a_3$ form a blocking pair. This example shows that player $p_1$ must identify the gap among the first $\sigma_i(m_i^*)$ ranked arms to find a stable matching in the market, which further illustrates the crucial role of $\text{gap}_4$ in learning the true stable matching. We leave the lower bound in general markets as an important future direction.

## 5  Experiments

In this section, we compare our Algorithm 1 (abbreviated as AETGS-E) with baselines ETGS [16], ML-ETC [31] and Phased ETC [4] which also enjoy guarantees for player-optimal stable regret in general decentralized one-to-one markets. To better illustrate the advantages of our algorithm, especially when $N$ is much smaller than $K$, we set $N = 3$ and $K = 10$. The preference rankings for both players and arms are generated as random permutations. The preference gap between any adjacent ranked arms is set as 0.1. The feedback $X_{i,j}(t)$ for player $p_i$ on arm $a_j$ at time $t$ is drawn independently from the Gaussian distribution with mean $\mu_{i,j}$ and variance 1. We report the maximum cumulative player-optimal stable regret among all players and the cumulative player-optimal instability in Figure 1 (a) and (b), respectively. Here the cumulative player-optimal unstability is defined as the number of matchings that are not the player-optimal stable one. All algorithms run for $T = 100k$ rounds and all results are averaged over 50 independent runs. The error bars represent standard errors, which are computed as standard deviations divided by $\sqrt{50}$.

As shown in the figure, our AETGS-E algorithm, which only conducts necessary explorations over unknown preferences and promptly eliminates sub-optimal arms, achieves the least cumulative regret and cumulative player-optimal unstability among all baselines. The ML-ETC and ETGS algorithms need to sufficiently explore $K$ arms to estimate the full preference ranking, requiring more exploration time to find the player-optimal stable matching. The PhasedETC algorithm has not yet converged within the displayed rounds due to the cold start problem.

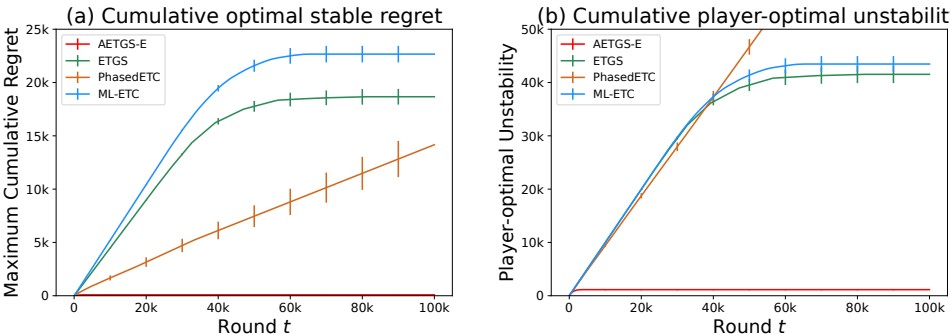

Figure 1: Experimental comparisons of our AETGS-E with ETGS, ML-ETC and Phased ETC in one-to-one decentralized markets with $N = 3$ players and $K = 10$ arms.

## 6 Centralized UCB Algorithm for Markets with $\alpha$-condition

In this section, we provide a new analysis for the centralized UCB algorithm in markets satisfying $\alpha$-condition. The algorithm is first introduced by Liu *et al.* [20]. For completeness, we present the full algorithm in Algorithm 2. At each round, players submit their UCB rankings to the centralized platform (Line 3). The platform runs the GS algorithm (based on players' submitted rankings) and returns the partner to each player (Line 4).

---

**Algorithm 2** centralized UCB

**Input:** $N, K$.
1: Initialize: $\hat{\mu}_{i,j}(0) = 0, T_{i,j}(0) = 0, \text{UCB}_{i,j}(1) = \infty, \forall i \in [N], j \in [K]$.
2: **for** round $t = 1, 2, \ldots, T$ **do**
3:     Receive rankings $\hat{\sigma} := \{\hat{\sigma}_i\}_{i\in[N]}$ according to the decreasing order of $\{\text{UCB}_{i,j}(t)\}_{j\in[K]}, \forall i \in [N]$;
4:     $A_i(t) \leftarrow$ **Gale-Shapley** $(\hat{\sigma}, \pi)$ for each player $p_i$;
5:     Observe $X_i(t)$, and update $\hat{\mu}_{i,j}(t), T_{i,j}(t), \text{UCB}_{i,j}(t+1)$ for each $p_i, a_j$;
6: **end for**

---

In the following, we introduce the $\alpha$-condition. Conditions guaranteeing the unique stable matching have been widely studied in the offline setting [5, 13] and also the online setting to improve the learning efficiency [28, 4, 22]. Among these conditions, the $\alpha$-condition is shown to be the weakest sufficient one [13] and incorporate the conditions studied in existing works [28, 4, 22].

Let $\beta$ denote a pair of permutations of $[N]$ and $[K]$. Then $[N]_\beta = \{Q_1^{(\beta)}, \ldots, Q_N^{(\beta)}\}$ and $[K]_\beta = \{q_1^{(\beta)}, \ldots, q_K^{(\beta)}\}$ denote permutations of the ordered sets $[N]$ and $[K]$, respectively. The $j$-th player in $[N]_\beta$ is the $Q_j^{(\beta)}$-th player in $[N]$, and the $k$-th arm in $[K]_\beta$ is the $q_k^{(\beta)}$-th arm in $[K]$. Then we can define the $\alpha$-condition below.

**Definition 6.1.** The $\alpha$-condition is satisfied if there is a stable matching $(\mathbf{j}^*, \mathbf{i}^*)$, a left-order of players and arms s.t. $\forall i \in [N]_l, \forall j > i, j \in [K]_l : \mu_{i,j_i^*} > \mu_{i,j}$ where $j_i^*$ is the partner of player $p_i$ in stable matching $(\mathbf{j}^*, \mathbf{i}^*)$, and a (possibly different) right-order of players and arms s.t. $\forall j < i \leq N, q_j \in [K]_r, Q_i \in [N]_r : \pi_{q_j, Q_{i_{q_j}^*}} > \pi_{q_j, Q_i}$. Here similarly, $i_{q_j}^*$ is the partner of arm $a_{q_j}$ in stable matching $(\mathbf{j}^*, \mathbf{i}^*)$.

Without loss of generality, we consider the identity of players and arms is just the left order, i.e., $[N] = [N]_l$ and $[K] = [K]_l$. Thus we only deal with player order $Q_i^{(r)} = Q_i$ and arm order $q_j^{(r)} = q_j$, for $i \in [N], j \in [K]$ in the rest of the paper. Under $\alpha$-condition, it is easy to inductively verify that for any $i \in [N]$, the player $p_i$ is matched with arm $a_i$, and the player $p_{Q_i}$ is matched with the arm $a_{q_i}$ in the unique stable matching [4].

## 6.1 Theoretical Results

We analyze the regret for the centralized UCB algorithm under $\alpha$-condition.

**Theorem 6.2.** *When preferences of participants satisfy $\alpha$-condition, following Algorithm 2, the stable regret for each player $p_i$ satisfies*

$$Reg_i(T) \leq O\left(N\log T/\Delta_N^2 + K\log T/\Delta\right).$$

The centralized UCB algorithm is proposed by Liu *et al.* [20] and shown to have $O(NK\log T/\Delta^2)$ player-pessimal stable regret for general markets. We provide a new analysis for markets satisfying $\alpha$-condition which removes the dependence of $K$ in the regret. Due to the space limit, we discuss the key idea of the proof below and defer the detailed proof to Appendix C.

We investigate the preference structure of $\alpha$-condition to obtain the improved analysis. For player $p_i$, its regret is due to selecting sub-optimal arm $a_k$ with $\mu_{i,k} < \mu_{i,i}$. Arm $a_k$ will be selected by $p_i$ when its UCB value is higher than $p_i$'s stable matched arm $a_i$, which time is bounded by $O(\log T/\Delta_{i,i,k}^2)$, and one $\Delta_{i,i,k}$ on the denominator can be eliminated when multiplying $\Delta_{i,i,k}$ to compute regret. This contributes $O(K\log T/\Delta)$ regret since there are at most $K-1$ sub-optimal arms. It is worth noting that arm $a_k$ will also be selected by $p_i$ if $p_i$ is rejected by $a_i$ in the GS algorithm. Recall that under $\alpha$-condition, there is a right order $Q_{i'} = i \in [N]_r$ for player $p_i$, such that $\forall i'' > i', Q_{i''} \in [N]_r : \pi_{q_{i'},Q_{i'}} > \pi_{q_{i'},Q_{i''}}$, which means arm $a_{q_{i'}} = a_i$ can only prefer players $p_{Q_1}, p_{Q_2}, \cdots, p_{Q_{i'-1}}$ than player $p_{Q_{i'}} = p_i$. Thus $p_i$ is rejected by $a_i$ only when these players select $a_i$, and $a_i$ is sub-optimal for those players. To bound the regret of $p_i$ when being rejected, we just need to bound the exploration times of these players $p_{Q_1}, p_{Q_2}, \cdots, p_{Q_{i'-1}}$ on arm $a_i$. However, the exploration time of a single player $p_{Q_\ell}$ with $1 \leq \ell \leq i'-1$ on $a_i$ can not be trivially bounded by $O(\log T/\Delta_N^2)$ since $p_{Q_\ell}$ may have to select arm $a_i$ after rejected by its stable arm $a_{q_\ell}$ in offline GS, where $a_{q_\ell}$ might be selected by $p_{Q_1}, \cdots, p_{Q_{\ell-1}}$. This leads to a recursion form. We control this term using the fact that when a player is rejected by its stable matched arm in the GS, it can date back to a higher right-order player wrongly over-estimate its preference for a sub-optimal arm. This key observation and the definition of $\Delta$ make it possible to derive the final $O\left(N\log T/\Delta_N^2\right)$ bound.

## 7 Conclusion

In this paper, we investigate the problem of whether a tighter bound can be derived for the bandit learning problem in two-sided matching markets. For the general one-to-one matching markets, we try to improve the learning efficiency of the existing algorithms. By integrating the offline GS procedure into the online learning process and carefully designing the elimination strategy, we show that the player-optimal stable regret can be upper bounded by $O(N^2\log T/\Delta^2 + K\log T/\Delta)$. This result removes the dependence on $K$ in the main order term of existing works and improves the state-of-the-art result [31, 16] in common cases where the number of players is much smaller than that of arms. An experiment is conducted to verify its advantage over other baselines in such markets. We also present a novel analysis for the centralized UCB algorithm in markets satisfying $\alpha$-condition and derive an improved $O(N\log T/\Delta_N^2 + K\log T/\Delta)$ regret upper bound.

One significant future direction is to investigate the optimality of algorithms. Although the dependence on $N, K, T$ in Theorem 6.2 matches the lower bound, the definition of $\Delta$ differs. It remains unclear how the upper bound changes with the same $\Delta$. Furthermore, since the lower bound provided by [28] applies only to special markets, and the learning problem in general markets is more challenging due to the complex preference structure, determining whether an algorithm can perform better in general markets is still an open problem.

## Acknowledgments and Disclosure of Funding

The corresponding author Shuai Li is supported by National Key Research and Development Program of China (2022ZD0114804) and National Natural Science Foundation of China (62376154). Fang Kong is supported by the Baidu Scholarship.

We thank Yuhao Zhang and Wenqian Wang for valuable discussions and suggestions on the proof of Lemma B.2.

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

# A  Proof of Theorem 4.1

Define $\mathcal{F} = \left\{ \exists 1 \le t \le T, i \in [N], j \in [K] : |\hat{\mu}_{i,j}(t) - \mu_{i,j}| > \sqrt{\frac{6 \log T}{T_{i,j}(t)}} \right\}$ as the failure event that the estimated reward is far from the expected reward at some time and some player-arm pair. The regret can be upper bounded as follows.

$$
\begin{aligned}
Reg_i(t) =& \mathbb{E}\left[ \sum_{t=1}^{T} \left( \mu_{i,m_i^*} - X_i(t) \right) \right] \\
\le& \mathbb{E}\left[ \sum_{t=1}^{T} \left( \mu_{i,m_i^*} - X_i(t) \right) | \neg \mathcal{F} \right] + \mathbb{P}\left(\mathcal{F}\right) \cdot T \cdot \mu_{i,m_i^*} \\
\le& \mathbb{E}\left[ \sum_{t=1}^{T} \sum_{a_j} \mathbb{1}\left\{ \bar{A}_i(t) = a_j \right\} \cdot \Delta_{i,m_i^*,j} | \neg \mathcal{F} \right] + \mathbb{E}\left[ \sum_{t=1}^{T} \mathbb{1}\left\{ \bar{A}_i(t) = \emptyset \right\} \cdot \mu_{i,m_i^*} | \neg \mathcal{F} \right] \\
& + \mathbb{P}\left(\mathcal{F}\right) \cdot T \cdot \mu_{i,m_i^*} \\
\le& 96N^2 \log T / \Delta^2 + 96K \log T / \Delta + 192 N^2 \log T / \Delta^2 + 2NK \qquad (1) \\
=& O\left( N^2 \log T / \Delta^2 + K \log T / \Delta \right).
\end{aligned}
$$

where Eq. (1) holds based on Lemma A.1, A.2, and A.4.

**Lemma A.1.**

$$
\mathbb{E}\left[ \sum_{t=1}^{T} \sum_{a_j} \mathbb{1}\left\{ \bar{A}_i(t) = a_j \right\} \cdot \Delta_{i,m_i^*,j} | \neg \mathcal{F} \right] \le 96N^2 \log T / \Delta^2 + 96K \log T / \Delta.
$$

*Proof.* Recall that player $p_i$ would update the available set $\mathcal{A}_i$ when other players $p_{i'}$ sets $\mathrm{E}_{i'} =$ False and $\pi_{A_{i'},i'} > \pi_{A_{i'},i}$ as Line 14. Denote $t_s$ as the round index when this operation happens for the $s$-th time. Without loss of generality, let $t_0 = 1$.

Recall that at a high level, each time another player $p_{i'}$ sets $\mathrm{E}_{i'}$ as False, it means that $p_{i'}$ learns its most preferred arm in current available set. Combined with $\mathcal{F}$ and Lemma A.6, the determined arm of players during each exploration would be truly their most preferred one. Thus the AETGS algorithm is an online version of GS and the $s'$-th time player $p_{i'}$ sets $\mathrm{E}_{i'}$ as False is equivalent to that $p_{i'}$ proposes its $s'$-th most preferred arm in the offline GS. According to Lemma A.3, at most $N - 1$ arms are proposed by all players before reaching stability. Thus for player $p_i$, the operation in Line 14 would happen for at most $N - 1$ times.

Recall that for each $s$, during time $t_s$ to $t_{s+1}$, player $p_i$ would explore all available arms in a round-robin manner, eliminate sub-optimal arms until $N$ arms are in the set, and focuses on the best one among these $N$ when it is identified. For convenience, denote $R_s$ as the set of the remaining $N$ arms that $p_i$ explored in $\mathcal{A}_i$ in a round-robin manner until condition Line 9 is satisfied, $D_s$ as the set of arms that $p_i$ eliminated due to condition Line 6, and $j_s$ as the arm that $p_i$ focuses from the time it sets $\mathrm{E}_i$ as False to time $t_{s+1} - 1$. Then it holds that

$$
\begin{aligned}
& \mathbb{E}\left[ \sum_{t=1}^{T} \sum_{a_j} \mathbb{1}\left\{ \bar{A}_i(t) = a_j \right\} \cdot \Delta_{i,m_i^*,j} | \neg \mathcal{F} \right] \\
\le& \mathbb{E}\left[ \sum_{s=0}^{N-1} \sum_{t=t_s}^{t_{s+1}-1} \sum_{a_j} \mathbb{1}\left\{ \bar{A}_i(t) = a_j \right\} \cdot \Delta_{i,m_i^*,j} | \neg \mathcal{F} \right] \\
\le& \mathbb{E}\left[ \sum_{s=0}^{N-1} \sum_{t=t_s}^{t_{s+1}-1} \left( \sum_{a_j \in R_s} \mathbb{1}\left\{ \bar{A}_i(t) = a_j \right\} \cdot \Delta_{i,m_i^*,j} + \sum_{a_j \in D_s} \mathbb{1}\left\{ \bar{A}_i(t) = a_j \right\} \cdot \Delta_{i,m_i^*,j} \right. \right. \\
& \left. \left. + \mathbb{1}\left\{ \bar{A}_i(t) = a_{j_s} \right\} \cdot \Delta_{i,m_i^*,j_s} \right) | \neg \mathcal{F} \right]
\end{aligned}
$$

$$\leq \mathbb{E}\left[\sum_{s=0}^{N-1}\sum_{t=t_s}^{t_{s+1}-1}\left(\sum_{a_j\in R_s}\mathbb{1}\{\bar{A}_i(t)=a_j\}\cdot\Delta_{i,m_i^*,j}+\sum_{a_j\in D_s}\mathbb{1}\{\bar{A}_i(t)=a_j\}\cdot\Delta_{i,m_i^*,j}\right)\;|\neg\mathcal{F}\right],$$

(2)

where Eq. (2) is due to that, based on Lemma A.6 and the offline GS algorithm, the arm $j_s$ before GS stops would be better than $m_i^*$, thus the regret caused by selecting these arms is less than 0.

For the first term in Eq. (2), we have

$$\mathbb{E}\left[\sum_{s=0}^{N-1}\sum_{t=t_s}^{t_{s+1}-1}\sum_{a_j\in R_s}\mathbb{1}\{\bar{A}_i(t)=a_j\}\cdot\Delta_{i,m_i^*,j}\;|\neg\mathcal{F}\right]$$

(3)

$$=\mathbb{E}\left[\sum_{s=0}^{N-1}\sum_{a_j\in R_s}\sum_{t=t_s}^{t_{s+1}-1}\mathbb{1}\{\bar{A}_i(t)=a_j\}\cdot\Delta_{i,m_i^*,j}\;|\neg\mathcal{F}\right]$$

$$\leq\mathbb{E}\left[\sum_{s=0}^{N-1}\sum_{a_j\in R_s}\frac{96\log T}{\Delta_{i,j_s,j_{s+1}}^2}\cdot\Delta_{i,m_i^*,j}\;|\neg\mathcal{F}\right]$$

(4)

$$\leq\frac{96N^2\log T}{\Delta^2}.$$

(5)

where Eq. (4) is due to Lemma A.5, Eq. (5) holds since $R_s$ contains no more than $N$ arms according to the elimination condition (Line 6).

We now analyze the second term in Eq. (2). For any arm $a_j$ and $s\in\{0,...,N-1\}$, denote $T_{i,j,s}$ as the value of $T_{i,j}$ at the end of the round $t_{s+1}-1$. For $s\geq 1$ and arm $a_j\in D_s$, if $T_{i,j,s-1}\leq 96\log T/\Delta_{i,j_{s-1},j}^2$, it must hold that $T_{i,j,s}:=\sum_{s'\leq s}(T_{i,j,s'}-T_{i,j,s'-1})\leq 96\log T/\Delta_{i,j_s,j}^2$ to ensure arm $a_j$ is eliminated from $\mathcal{A}_i$ at step $s$ based on Lemma A.5. On the other hand, if $T_{i,j,s-1}>96\log T/\Delta_{i,j_{s-1},j}^2$, based on Lemma A.5, it holds that $T_{i,j,s}-T_{i,j,s-1}\leq 96\log T/\Delta_{i,j_s,j}^2-96\log T/\Delta_{i,j_{s-1},j}^2$ when $a_j$ is eliminated. For any arm $a_j$, denote $s_{j,1}:=\max_{0\leq s\leq N-1}\left\{s:T_{i,j,s}\leq 96\log T/\Delta_{i,j_s,j}^2\right\}$ as the last step when the number of observation times on $a_j$ is less than that threshold. Then the second term in Eq. (2) satisfies

$$\mathbb{E}\left[\sum_{s=0}^{N-1}\sum_{t=t_s}^{t_{s+1}-1}\sum_{a_j\in D_s}\mathbb{1}\{\bar{A}_i(t)=a_j\}\cdot\Delta_{i,m_i^*,j}\;|\neg\mathcal{F}\right]$$

$$=\mathbb{E}\left[\sum_{a_j\in\mathcal{K}}\sum_{s:a_j\in D_s}\sum_{t=t_s}^{t_{s+1}-1}\mathbb{1}\{\bar{A}_i(t)=a_j\}\cdot\Delta_{i,m_i^*,j}\;|\neg\mathcal{F}\right]$$

$$=\mathbb{E}\left[\sum_{a_j\in\mathcal{K}}\sum_{s:a_j\in D_s}(T_{i,j,s}-T_{i,j,s-1})\cdot\Delta_{i,m_i^*,j}\;|\neg\mathcal{F}\right]$$

$$=\mathbb{E}\left[\sum_{a_j\in\mathcal{K}}\left(\sum_{s:a_j\in D_s,s\leq s_{j,1}}(T_{i,j,s}-T_{i,j,s-1})+\sum_{s:a_j\in D_s,s>s_{j,1}}(T_{i,j,s}-T_{i,j,s-1})\right)\cdot\Delta_{i,m_i^*,j}\;|\neg\mathcal{F}\right]$$

$$\leq\mathbb{E}\left[\sum_{a_j\in\mathcal{K}}\left(T_{i,j,s_{j,1}}+\sum_{s:a_j\in D_s,s>s_{j,1}}(T_{i,j,s}-T_{i,j,s-1})\right)\cdot\Delta_{i,m_i^*,j}\;|\neg\mathcal{F}\right]$$

$$\leq\sum_{a_j\in\mathcal{K}}\left(\frac{96\log T}{\Delta_{i,j_{s_{j,1}},j}^2}+\sum_{s:a_j\in D_s,s>s_{j,1}}\left(96\log T/\Delta_{i,j_s,j}^2-96\log T/\Delta_{i,j_{s-1},j}^2\right)\right)\cdot\Delta_{i,m_i^*,j}$$

$$\le \sum_{a_j \in \mathcal{K}} \frac{96 \log T}{\Delta_{i,m_i^*,j}^2} \cdot \Delta_{i,m_i^*,j} \le 96K \log T/\Delta \,,$$

where the second last line is due to the definition of $s_{j,1}$ and the above analysis.

Above all,

$$\mathbb{E}\left[\sum_{t=1}^{T} \sum_{a_j} \mathbb{1}\left\{\bar{A}_i(t) = a_j\right\} \cdot \Delta_{i,m_i^*,j} \mid \neg\mathcal{F}\right] \le \text{Eq. (2)} \le 96N^2 \log T/\Delta^2 + 96K \log T/\Delta \,.$$

$\square$

**Lemma A.2.**

$$\mathbb{E}\left[\sum_{t=1}^{T} \mathbb{1}\left\{\bar{A}_i(t) = \emptyset\right\} \cdot \mu_{i,m_i^*} \mid \neg\mathcal{F}\right] \le 192N^2 \log T/\Delta^2 \,.$$

*Proof.* Based on the AETGS algorithm, when $\mathrm{E}_i = \text{True}$, the central platform would assign arms in $\mathcal{A}_i$ to player $p_i$ in a round-robin manner. Since the number of arms $|\cup_{i:\mathrm{E}_i=\text{True}}\mathcal{A}_i|$ to be explored is larger than the number $\sum_i \mathbb{1}\{\mathrm{E}_i = \text{True}\}$ of players with $\mathrm{E}_i = \text{True}$ based on the elimination condition in Line 6, we can assume that there is no collision in the exploration phase as discussed in Section 4. So the regret caused by collision only occurs during time with $\mathrm{E}_i = \text{False}$.

Denote $\underline{t}_s$ and $\bar{t}_s$ as the round index when $p_i$ sets $\mathrm{E}_i$ as False for the $s$-th time and as True for the $s+1$-th time, respectively. Recall that when $\mathrm{E}_i = \text{False}$, $p_i$ will always select arm $A_i$. Here we use $j_s$ to represent the arm that is selected by $p_i$ from time $\underline{t}_s$ to $\bar{t}_s$.

Further, recall that in the AETGS algorithm, each time an arm is added into $\mathcal{D}_i$ (Line 13), the eliminated arms may be contained into $\mathcal{A}_i$ again. And only when other players focus on their currently most preferred arm, such operation of adding arms to $\mathcal{D}$ happens. Based on Lemma A.3, such an operation happens for at most $N$ times. For any player $p_{i'}$, denote $t_{i',r}$ as the round index when $p_{i'}$ adds arms to $\mathcal{D}_{i'}$ (Line 13) for $r$-th time. Then $\{t_{i',r}\}_{r\in[N]}$ further divide $\{[\underline{t}_s, \bar{t}_s]\}_{s\in[N]}$ into at most $2N$ slices. We use $\underline{t}'_s, \bar{t}'_s$ to represent the start round and end round index of the $s$-th slice, where $s \in [2N]$. Based on Lemma A.5, $p_{i'}$ and $p_i$ would select the same arm for at most $96 \log T/\Delta^2$ times within each slice.

Then the regret satisfies

$$\mathbb{E}\left[\sum_{t=1}^{T} \mathbb{1}\left\{\bar{A}_i(t) = \emptyset\right\} \cdot \mu_{i,m_i^*} \mid \neg\mathcal{F}\right] \le \mathbb{E}\left[\sum_{s=1}^{N} \sum_{t=\underline{t}_s}^{\bar{t}_s} \mathbb{1}\left\{\bar{A}_i(t) = \emptyset\right\} \cdot \mu_{i,m_i^*} \mid \neg\mathcal{F}\right]$$

$$= \mathbb{E}\left[\sum_{s=1}^{N} \sum_{t=\underline{t}_s}^{\bar{t}_s} \mathbb{1}\left\{\bar{A}_i(t) = \emptyset, A_i(t) = j_s\right\} \cdot \mu_{i,m_i^*} \mid \neg\mathcal{F}\right]$$

$$\le \mathbb{E}\left[\sum_{i'\neq i} \sum_{s=1}^{N} \sum_{t=\underline{t}_s}^{\bar{t}_s} \mathbb{1}\left\{A_i(t) = A_{i'}(t) = j_s\right\} \cdot \mu_{i,m_i^*} \mid \neg\mathcal{F}\right]$$

$$\le \mathbb{E}\left[\sum_{i'\neq i} \sum_{s=1}^{2N} \sum_{t=\underline{t}'_s}^{\bar{t}'_s} \mathbb{1}\left\{A_i(t) = A_{i'}(t)\right\} \cdot \mu_{i,m_i^*} \mid \neg\mathcal{F}\right]$$

$$\le \sum_{i'\neq i} \sum_{s=1}^{2N} 96 \log T/\Delta^2 \cdot \mu_{i,m_i^*}$$

$$\le 192N^2 \log T/\Delta^2 \,.$$

$\square$

**Lemma A.3.** *In the offline GS algorithm, at most $N-1$ arms have been proposed by players before the algorithm stops.*

*Proof.* Based on the offline GS algorithm, once an arm is proposed, it has a temporary player. By contradiction, once $N$ arms have been proposed, it means that $N$ players are occupied. In this case, each player has a partner and the algorithm stops. $\qquad\square$

**Lemma A.4.**
$$\mathbb{P}\left(\mathcal{F}\right) \leq 2NK/T.$$

*Proof.*

$$\mathbb{P}\left(\mathcal{F}\right) = \mathbb{P}\left(\exists 1 \leq t \leq T, i \in [N], j \in [K] : |\hat{\mu}_{i,j}(t) - \mu_{i,j}| > \sqrt{\frac{6\log T}{T_{i,j}(t)}}\right)$$

$$\leq \sum_{t=1}^{T} \sum_{i \in [N]} \sum_{j \in [K]} \mathbb{P}\left(|\hat{\mu}_{i,j}(t) - \mu_{i,j}| > \sqrt{\frac{6\log T}{T_{i,j}(t)}}\right)$$

$$\leq \sum_{t=1}^{T} \sum_{i \in [N]} \sum_{j \in [K]} \sum_{s=1}^{t} \mathbb{P}\left(T_{i,j}(t) = s, |\hat{\mu}_{i,j}(t) - \mu_{i,j}| > \sqrt{\frac{6\log T}{s}}\right)$$

$$\leq \sum_{t=1}^{T} \sum_{i \in [N]} \sum_{j \in [K]} t \cdot 2\exp(-3\ln T)$$

$$\leq 2NK/T,$$

where the second last inequality is due to Lemma B.1. $\qquad\square$

**Lemma A.5.** *For any player $p_i$, let $\bar{T}_i = 96\log T/\Delta^2$. For any two arms $j, j'$ with $\mu_{i,j} > \mu_{i,j'}$ and $\sigma_i(a_j) \in [1, \sigma_i(m_i^*)]$, if $T_i(t) := \min\{T_{i,j}(t), T_{i,j'}(t)\} > \bar{T}_i$, we have $\mathrm{UCB}_{i,j'}(t) < \mathrm{LCB}_{i,j}(t)$ conditioned on $\neg\mathcal{F}$.*

*Proof.* By contradiction, suppose $\mathrm{UCB}_{i,j'}(t) \geq \mathrm{LCB}_{i,j}(t)$. According to $\neg\mathcal{F}$ and the definition of LCB and UCB, we have

$$\mu_{i,j} - 2\sqrt{\frac{6\log T}{T_i(t)}} \leq \mathrm{LCB}_{i,j}(t) \leq \mathrm{UCB}_{i,j'}(t) \leq \mu_{i,j'} + 2\sqrt{\frac{6\log T}{T_i(t)}}.$$

We can then conclude $\Delta_{i,j,j'} = \mu_{i,j} - \mu_{i,j'} \leq 4\sqrt{\frac{6\log T}{T_i(t)}}$, which implies that $T_i(t) \leq \frac{96\log T}{\Delta_{i,j,j'}^2} \leq \frac{96\log T}{\Delta^2}$. This contradicts the fact that $T_i(t) > \bar{T}_i$. $\qquad\square$

**Lemma A.6.** *Conditioned on $\neg\mathcal{F}$, at any time $t$, $\mathrm{UCB}_{i,j}(t) < \mathrm{LCB}_{i,j'}(t)$ implies $\mu_{i,j} < \mu_{i,j'}$.*

*Proof.* According to the definition of LCB and UCB, we have

$$\mathrm{LCB}_{i,j}(t) = \hat{\mu}_{i,j}(t) - \sqrt{\frac{6\log T}{T_{i,j}(t)}} \leq \mu_{i,j} \leq \hat{\mu}_{i,j}(t) + \sqrt{\frac{6\log T}{T_{i,j}(t)}} = \mathrm{UCB}_{i,j}(t),$$

where two inequalities comes from $\neg\mathcal{F}$. Thus if $\mathrm{UCB}_{i,j}(t) < \mathrm{LCB}_{i,j'}(t)$, there would be

$$\mu_{i,j} \leq \mathrm{UCB}_{i,j}(t) < \mathrm{LCB}_{i,j'}(t) \leq \mu_{i,j'}.$$

The lemma can thus be proved. $\qquad\square$

# B  Technical Lemmas

**Lemma B.1.** *(Corollary 5.5 in Lattimore and Szepesvári [19]) Assume that $X_1, X_2, \ldots, X_n$ are independent, $\sigma$-subgaussian random variables centered around $\mu$. Then for any $\varepsilon > 0$,*

$$\mathbb{P}\left(\frac{1}{n}\sum_{i=1}^{n} X_i \geq \mu + \varepsilon\right) \leq \exp\left(-\frac{n\varepsilon^2}{2\sigma^2}\right), \quad \mathbb{P}\left(\frac{1}{n}\sum_{i=1}^{n} X_i \leq \mu - \varepsilon\right) \leq \exp\left(-\frac{n\varepsilon^2}{2\sigma^2}\right).$$

**Lemma B.2.** *(Arrangement of players' round-robin exploration) Suppose there are $N$ players who need to explore their respective $N$ arms. There exists an assignment such that during $2N$ rounds, each player can match with each of its arm for once.*

*Proof.* Without loss of generality, let's assume that players assign their respective $N$ arms in $2N$ rounds one by one, based on the players' and arms' indices, aiming to ensure that no arm is assigned to more than one player at the same round. By contradiction, suppose when player $p_i$ assigns its $j$-th arm, there is no available round to make this assignment due to conflicting constraints. Given that player $p_i$ is currently assigning the $j$-th arm, it implies that there are $2N - j + 1$ rounds where no arm is assigned to player $p_i$. Since none of these rounds satisfy the conflict constraint, it means that the previous $i - 1$ players assigned arm $j$ in these $2N - j + 1$ rounds. This creates a contradiction since the first $i - 1$ players can only occupy $i - 1$ rounds when selecting arm $j$, where $i - 1 < 2N - j + 1$ with $i \leq N, j \leq N$. □

## C   Proof of Theorem 6.2

In this section, we analyze the regret of the centralized-UCB algorithm under $\alpha$-condition. Recall that $\mathcal{F} = \left\{ \exists 1 \leq t \leq T, i \in [N], j \in [K] : |\hat{\mu}_{i,j}(t) - \mu_{i,j}| > \sqrt{\frac{6 \log T}{T_{i,j}(t)}} \right\}$ is the failure event that the estimated reward is far from the expected reward at some time and some player-arm pair.

For any player $p_i$ with $i \in [N]$, we know that its stable arm is $a_i$ under $\alpha$-condition. Thus its regret can be decomposed as

$$Reg_i(T) \leq \mathbb{E}\left[ \sum_{k:\mu_{i,k} < \mu_{i,i}} \Delta_{i,i,k} \sum_{t=1}^{T} \mathbb{1}\{\bar{A}_i(t) = k, \neg\mathcal{F}\} \right] + T \cdot \mathbb{P}(\mathcal{F}).$$

The first term is the number of selections for sub-optimal arms. The second term is the regret caused by the bad events.

For the arm $a_k$ such that it is sub-optimal for player $p_i$, i.e., $\mu_{i,k} < \mu_{i,i}$, it will be selected because the preference for arm $a_k$ of player $p_i$ is estimated higher than its stable matched arm $a_i$, or player $p_i$ is rejected by arm $a_k$ in the GS algorithm. Note that under $\alpha$-condition, there is a right order $Q_{i'} = i \in [N]_r$ for player $p_i$, such that $\forall i' < i'' \leq N, Q_{i''} \in [N]_r : \pi_{q_{i'}, Q_{i'}} > \pi_{q_{i'}, Q_{i''}}$, which means arm $a_{q_{i'}} = a_i$ can only prefer players $p_{Q_1}, p_{Q_2}, \cdots, p_{Q_{i'-1}}$ than player $p_{Q_{i'}} = p_i$. Denote The right-order mapping for $\alpha$-condition for player $p_i$ is $lr(i)$ so that $Q_{lr(i)} = i$ with $Q_i$ defined in Definition 6.1, and $lr(i) \leq N$. For player $p_i$, denote $\mathcal{G}_{t,i} := \{\forall 1 \leq i' \leq lr(i) - 1, \bar{A}_{Q_{i'}}(t) \neq i\}$ as the event all players preferred by arm $a_i$ do not select $p_i$ at time $t$. Then the number of selections for sub-optimal arm $a_k$ can be decomposed as

$$\mathbb{E}\left[ \sum_{t=1}^{T} \mathbb{1}\{\bar{A}_i(t) = k, \neg\mathcal{F}\} \right]$$

$$= \mathbb{E}\left[ \sum_{t=1}^{T} \mathbb{1}\{\bar{A}_i(t) = k, \mathcal{G}_{t,i}, \neg\mathcal{F}\} \right] + \mathbb{E}\left[ \sum_{t=1}^{T} \mathbb{1}\{\bar{A}_i(t) = k, \neg\mathcal{G}_{t,i}, \neg\mathcal{F}\} \right]$$

$$\leq \mathbb{E}\left[ \sum_{t=1}^{T} \mathbb{1}\{\bar{A}_i(t) = k, \text{UCB}_{i,k}(t) > \text{UCB}_{i,i}(t), \neg\mathcal{F}\} \right] + \mathbb{E}\left[ \sum_{t=1}^{T} \mathbb{1}\{\bar{A}_i(t) = k, \neg\mathcal{G}_{t,i}, \neg\mathcal{F}\} \right]$$

$$\leq \frac{24 \log T}{\Delta_{i,i,k}^2} + \mathbb{E}\left[ \sum_{t=1}^{T} \mathbb{1}\{\bar{A}_i(t) = k, \neg\mathcal{G}_{t,i}, \neg\mathcal{F}\} \right].$$

The last inequality is from Lemma C.1.

For the second term in the RHS of the last inequality, $\mathbb{E}\left[ \sum_{t=1}^{T} \mathbb{1}\{\bar{A}_i(t) = k, \neg\mathcal{G}_{t,i}, \neg\mathcal{F}\} \right]$, we can sum over all sub-optimal arms and it turns out to be

$$\mathbb{E}\left[ \sum_{k} \sum_{t=1}^{T} \mathbb{1}\{\bar{A}_i(t) = k, \neg\mathcal{G}_{t,i}, \neg\mathcal{F}\} \right]$$

$$\leq \mathbb{E}\left[\sum_{t=1}^{T}\mathbb{1}\left\{\neg\mathcal{G}_{t,i}, \neg\mathcal{F}\right\}\right]$$

$$\leq \mathbb{E}\left[\sum_{i'=1}^{lr(i)-1}\sum_{t=1}^{T}\mathbb{1}\left\{\bar{A}_{Q_{i'}}(t) = i, \neg\mathcal{F}\right\}\right]$$

$$\leq \sum_{u'=1}^{lr(i)-1}\sum_{u''=u'+1}^{lr(i)}\frac{24\log T}{\Delta_{Q_{u'},q_{u'},q_{u''}}^{2}}$$

$$\leq \sum_{u'=1}^{lr(i)-1}\sum_{k=1}^{lr(i)-u'}\frac{24\log T}{(k\Delta_N)^2}$$

$$\leq (lr(i)-1)\left(\sum_{k=1}^{lr(i)-u'}\frac{1}{k^2}\right)\frac{24\log T}{\Delta_N^2}$$

$$\leq (lr(i)-1)\frac{5\pi^2\log T}{\Delta_N^2},$$

where the third inequity is from the Lemma C.2. The fourth inequality is from the definition of $\Delta_N$.
Above all, the stable regret of player $i$ can be bounded by

$$Reg_i(T) \leq \mathbb{E}\left[\sum_{k:\mu_{i,k}<\mu_{i,i}}\Delta_{i,i,k}\sum_{t=1}^{T}\mathbb{1}\left\{\bar{A}_i(t) = k, \neg\mathcal{F}\right\}\right] + T\cdot\mathbb{P}(\mathcal{F})$$

$$\leq \sum_{k:\mu_{i,k}<\mu_{i,i}}\Delta_{i,i,k}\frac{24\log T}{\Delta_{i,i,k}^2} + \Delta_{i,i,k}(lr(i)-1)\frac{5\pi^2\log T}{\Delta_N^2} + 2NK$$

$$\leq \frac{24K\log T}{\Delta} + (lr(i)-1)\frac{5\pi^2\log T}{\Delta_N^2} + 2NK$$

$$\leq O\left(\frac{K\log T}{\Delta} + \frac{N\log T}{\Delta_N^2}\right),$$

where the second inequality is based on Lemma A.4.

**Lemma C.1.** *Conditioned on $\neg\mathcal{F}$, under the traditional single-player UCB algorithm with single player $p_i$, the expected number of times at which the UCB index of arm $a_{j'}$ exceeds that of the better arm $a_j$, is at most $24\log(T)/\Delta_{i,j,j'}^2$ by round $T$.*

*Proof.* Conditioned on $\neg\mathcal{F}$, for any $i, j, t$ we have,

$$\mu_{i,j} - \sqrt{\frac{6\log(T)}{T_{i,j}(t-1)}} < \hat{\mu}_{i,j}(t-1) < \mu_{i,j} + \sqrt{\frac{6\log(T)}{T_{i,j}(t-1)}}. \tag{6a}$$

Recall that the UCB index is:

$$\text{UCB}_{i,j}(t) = \hat{\mu}_{i,j}(t-1) + \sqrt{\frac{6\log(T)}{T_{i,j}(t-1)}}. \tag{6b}$$

The event that arm $a_{j'}$ is successfully selected for player $p_i$ rather than the better arm $a_j$ at time $t$ implies that

$$\text{UCB}_{i,j'}(t) > \text{UCB}_{i,j}(t). \tag{6c}$$

Hence,

$$\mu_{i,j'} + 2\sqrt{\frac{6\log(T)}{T_{i,j'}(t-1)}} \overset{(6a)}{>} \hat{\mu}_{i,j'}(t-1) + \sqrt{\frac{6\log(T)}{T_{i,j'}(t-1)}}$$

$$\overset{(6c)}{>} \hat{\mu}_{i,j}(t-1) + \sqrt{\frac{6\log(T)}{T_{i,j}(t-1)}}$$

$$> \mu_{i,j} - \sqrt{\frac{6\log(T)}{T_{i,j}(t-1)}} + \sqrt{\frac{6\log(T)}{T_{i,j}(t-1)}}$$

$$= \mu_{i,j} ,$$

which leads to

$$T_{i,j'}(t-1) < \frac{24\log(T)}{\Delta_{i,j,j'}^2} ,$$

where $\Delta_{i,j,j'}$ is the reward difference between the $\mu_{i,j'}$ and $\mu_{i,j}$.

$\square$

**Lemma C.2.** *For any player $p_i$ with right order $Q_{lr(i)}$, the following inequality holds:*

$$\mathbb{E}\left[\sum_{i'=1}^{lr(i)-1}\sum_{t=1}^{T}\mathbb{1}\big\{\bar{A}_{Q_{i'}}(t) = i, \neg\mathcal{F}\big\}\right]$$

$$\leq \mathbb{E}\left[\sum_{u'=1}^{lr(i)-1}\sum_{u''=u'+1}^{lr(i)}\sum_{t=1}^{T}\mathbb{1}\big\{\bar{A}_{Q_{u'}}(t) = q_{u''}, \mathcal{G}_{t,u'}, \neg\mathcal{F}\big\}\right]$$

$$\leq \sum_{u'=1}^{lr(i)-1}\sum_{u''=u'+1}^{lr(i)}\frac{24\log T}{\Delta_{Q_{u'},q_{u'},q_{u''}}^2} .$$

*Proof.* For player $p_i$ with right order $Q_{lr(i)}$, from $\alpha$-condition we have that its stable matched arm $a_i$ may prefer $p_{Q_1}, p_{Q_2}, \cdots, p_{Q_{lr(i)-1}}$ than player $p_{Q_{lr(i)}}$. For any $i' < lr(i)$, we know that the number of times player $p_{Q_{i'}}$ selects arm $a_i$ is decomposed as by

$$\mathbb{E}\left[\sum_{t=1}^{T}\mathbb{1}\big\{\bar{A}_{Q_{i'}}(t) = i, \neg\mathcal{F}\big\}\right]$$

$$= \mathbb{E}\left[\sum_{t=1}^{T}\mathbb{1}\big\{\bar{A}_{Q_{i'}}(t) = i, \mathcal{G}_{t,i'}, \neg\mathcal{F}\big\}\right] + \mathbb{E}\left[\sum_{t=1}^{T}\mathbb{1}\big\{\bar{A}_{Q_{i'}}(t) = i, \neg\mathcal{G}_{t,i'}, \neg\mathcal{F}\big\}\right] .$$

The event $\neg\mathcal{G}_{t,i'}$ implies that there exists another player $p_{Q_{i''}}$ with $i'' < i'$ that selects the stable arm of $p_{Q_{i'}}$. This leads to a recursion form. But it is easy to verify that every event $\neg\mathcal{G}_{t,i'}$ happens only when there exists two players $p_{Q_{u'}}, p_{Q_{u''}}$ with $u' < u'' \leq lr(i')$, such that player $p_{Q_{u'}}$ explores the stable matched arm $a_{q_{u''}}$ of $p_{Q_{u''}}$, i.e., $p_{Q_{u'}}$ selects $a_{q_{u''}}$ conditioned on $\mathcal{G}_{t,u'}$. And thus it holds that

$$\mathbb{E}\left[\sum_{i'=1}^{lr(i)-1}\sum_{t=1}^{T}\mathbb{1}\big\{\bar{A}_{Q_{i'}}(t) = i, \neg\mathcal{F}\big\}\right]$$

$$\leq \mathbb{E}\left[\sum_{u'=1}^{lr(i)-1}\sum_{u''=u'+1}^{lr(i)}\sum_{t=1}^{T}\mathbb{1}\big\{\bar{A}_{Q_{u'}}(t) = q_{u''}, \mathcal{G}_{t,u'}, \neg\mathcal{F}\big\}\right]$$

$$\leq \sum_{u'=1}^{lr(i)-1}\sum_{u''=u'+1}^{lr(i)}\frac{24\log T}{\Delta_{Q_{u'},q_{u'},q_{u''}}^2} .$$

$\square$

