# OpenReview forum: "Improved Analysis for Bandit Learning in Matching Markets"
_NeurIPS.cc/2024/Conference — NeurIPS 2024 poster_

### Official Review · Reviewer_Pjvv · 2024-06-28

**Soundness:** 3
**Presentation:** 3
**Contribution:** 2
**Rating:** 5
**Confidence:** 3

**Summary:**

This paper considers two-sided matching bandit problems, where the goal is to minimize regret against a player's optimal stable matching. The state-of-the-art algorithms achieve a regret bound of  $KlogT/\Delta^2$.  The authors suggest an algorithm using an adaptive online Gale-Shapley to achieve a regret bound of $N^2\log T/\Delta^2$ under $N<K$. Under $\alpha$-condition, they provide a new analysis for the previous algorithm to achieve a regret bound of $N\log T/\Delta^2$.

**Strengths:**

1. The authors suggest an adaptive online Gale-Shapley algorithm.
2. The suggested algorithms outperform the previous algorithms when $N$ is much smaller than $K$.
3. They demonstrate this result using synthetic experiments.

**Weaknesses:**

1. Without $\alpha$-condition, the upper bound has $N^2$ instead of $N$, which may not be tight with respect to $N$.
2. Even though they provide a new algorithm, it may be hard to see the technical novelty.

**Questions:**

Without $\alpha$-condition, is the lower bound linear with respect to $N$?

**Limitations:**

The authors have discussed the limitations in the conclusion.

---

> ### Author Rebuttal · Authors · 2024-08-07
>
> We thank the reviewer Pjvv for the valuable comments. Please find our response below.
>
> - $N^2$ regret for the general market
>
> In the matching market problem, each player needs to navigate both his individual explorations and game interactions with others. Considerable work has been done to investigate the optimal regret upper bound for this problem, starting from pessimal stable regret and unique stable regret to, only recently, results on player-optimal stable regret.
>
> Although the tightest regret bound remains unclear, our regret upper bound of \(O(N^2\log T/\Delta^2)\) eliminates the dependence on \(K\) in the main regret order term for the first time. This observation has enhanced our understanding of the role of the arm set size \(K\) in this learning problem and advances our comprehension of the core difficulty arising from the game interactions among $N$ players.
>
>
> - Technical novelty
>
> Dealing with individual players' exploration-exploitation trade-offs has been extensively studied in the literature. However, in matching markets, researchers have not swiftly comprehended how to integrate this trade-off with interactions among players. Previous efforts were devoted to addressing pessimal stable regret and uniqueness assumptions using classic UCB and TS strategies [4, 17, 19, 20, 21, 27]. Until recently, Zhang et al. (2022) [29] and Kong and Li (2023) [16] have found that ETC-type strategies better fit this problem. Nonetheless, their strategies distinctly separate individual exploration from multi-player interactions, requiring players to first estimate their full preferences and then perform exploitation. This may lead to over-exploration as some estimated preferences are not used in finding the stable matching.
>
> We propose a new perspective that integrates the players' learning process into GS steps. The algorithm promptly eliminates $K-N$ sub-optimal arms in each exploration cycle and adaptively switches between exploration and exploitation. Such design avoids unnecessary exploration of sub-optimal arms and successfully eliminates the dependence on $K$ in the regret, presenting a more suitable EE trade-off strategy in the matching market scenario.
>
>
> - Lower bound in the general market
>
> The state-of-the-art lower bound for this problem is $\Omega\left(\frac{N\log T}{\Delta^2}\right)$ under the serial dictatorship condition [27] where all arms have the same preferences.
> The general market may be more challenging since the game interactions among players become more complicated. In this setting, the specific dependence of the lower bound on $N$ and $K$ is still unclear. But as demonstrated by the result in [27], there must be at least a linear dependence on $N$. Investigating the optimality in general markets has been extensively studied and we leave this problem as an important future direction.

---

> > ### Comment · Reviewer_Pjvv · 2024-08-10
> > **Thank you for your response.**
> >
> > Overall, the novel approach is interesting because it achieves an $N^2$ dependency rather than $K$ in the setting without the alpha condition. However, the achieved regret bound may not be optimal, or the regret lower bound may not be tight enough. Also, I believe the technical novelty may not be significant enough. Therefore, I have decided to maintain my score.

---

### Official Review · Reviewer_Y8Bw · 2024-07-12

**Soundness:** 4
**Presentation:** 4
**Contribution:** 4
**Rating:** 7
**Confidence:** 3

**Summary:**

The paper studies the bandit learning problem in two-sided matching markets and provide improved regret analysis that nearly match the lower bound (although with slightly different definition of instance-dependent gaps). The bound is particularly useful when the number of players is much smaller than the number of arms. Numerical experiments show significant improvement under some special cases. The paper also improve the regret analysis for an existing UCB algorithm under an additional $\alpha$-condition.

**Strengths:**

1. The paper is very well-written. The flow is clear and easy-to-follow.
2. Strong technical novelty as well as solid theoretical results. The paper gives new insights on how to balance between exploration and exploitation in matching markets.
3. Numerical experiments show significant improvement compared to existing policies.

**Weaknesses:**

Although the dependence on $N$, $K$, $T$ is optimal, the main weakness lies in the difference of defining the instance-dependent gap $\Delta$.

**Questions:**

I would suggest the authors add a paragraph in the introduction to explcitly point out the contributions in this paper. In particular, I would like to see the main algorithmic design insight be emphasized from this work. For example, why previous algorithms fail to obtain low regret (due to over exploration)?

---

> ### Author Rebuttal · Authors · 2024-08-07
>
> We thank the reviewer Y8Bw for the valuable comments. Please find our response below.
>
>
> - Gap definition
>
> We agree that investigating the optimal dependence on the preference gap is an interesting future direction. Among all of the existing works, there are three types of gap definitions due to different learning processes to reach stability. Though it is still unclear whether the optimal result should be under different gap definitions, our provided Theorem 4.1 improves existing works for general markets in terms of both market sizes and gap definitions (without knowing $\Delta$). This result has further advanced our understanding of the EE trade-off problem in the matching market scenario.
>
> - Contribution of this work
>
> Thanks for your suggestion. We will add a paragraph summarizing our contribution in the introduction and the following is a draft:
>
> 1. We propose an adaptive online Gale-Shapley (AOGS) algorithm for general markets. State-of-the-art works [29, 16] explicitly separate the exploration and exploitation processes, which can lead to unnecessary regret, as exploring certain preference rankings may not contribute to the exploitation process. To avoid excessive exploration, our AOGS algorithm integrates the players’ learning process into the GS steps. Players explore only the necessary number of candidate arms and adaptively switch between exploration and exploitation.
>
>
> 2. We prove that the player-optimal stable regret of AOGS can be upper bounded by $O(N^2\log T/\Delta^2 + K\log T/\Delta)$. This is the first result that removes the dependence on $K$ in the main regret order term and improves existing works in common cases where $N$ is much smaller than $K$. We also conduct experiments to show the advantages of the algorithm.
>
>
> 3. We refine the analysis of the centralized UCB algorithm in [19] for markets satisfying the $\alpha$-condition. By investigating the preference hierarchy structure of the $\alpha$-condition, we demonstrate that the stable matching converges sequentially from player $1$ to player $N$. Through inductive analysis over $N$ players, we establish an $O(N \log T/\Delta^2 + K \log T/\Delta)$ regret upper bound, which enhances the original result for this algorithm in this specific market.

---

### Official Review · Reviewer_SWbh · 2024-07-14

**Soundness:** 3
**Presentation:** 3
**Contribution:** 2
**Rating:** 5
**Confidence:** 4

**Summary:**

The paper proposed a new algorithm for bandit learning in matching markets and showed new results on regret bounds. However, the combination of bandits with matching markets is wired. Is there any evidence that practitioners would like to use bandits for learning in markets? The novelty of the theory is also unclear.

**Strengths:**

The paper proposed a new algorithm for bandit learning in matching markets and showed new results on regret bounds.

**Weaknesses:**

The combination of bandits with matching markets is wired. Is there any evidence that practitioners would like to use bandits for learning in markets?

**Questions:**

1. The key novelty in the paper is unclear. What are the new techniques developed in the paper for proving the upper bound?

2. Do you have any empirical markets that could use the proposed algorithms?

3. How to generalize the techniques to decentralized markets?

**Limitations:**

The combination of bandits with matching markets is wired. Is there any evidence that practitioners would like to use bandits for learning in markets?

---

> ### Author Rebuttal · Authors · 2024-08-07
>
> We thank the reviewer SWbh for the valuable comments. Please find our response below.
>
> - Technical novelty
>
> Dealing with individual players' exploration-exploitation trade-offs has been extensively studied in the literature. However, in matching markets, researchers have not swiftly comprehended how to integrate this trade-off with interactions among players. Previous efforts were devoted to addressing pessimal stable regret and uniqueness assumptions using classic UCB and TS strategies [4, 17, 19, 20, 21, 27]. Until recently, Zhang et al. (2022) [29] and Kong and Li (2023) [16] have found that ETC-type strategies better fit this problem. Nonetheless, their strategies distinctly separate individual exploration from multi-player interactions, requiring players to first estimate their full preferences and then perform exploitation. This may lead to over-exploration as some estimated preferences are not used in finding the stable matching.
>
> We propose a new perspective that integrates the players' learning process into GS steps. The algorithm promptly eliminates $K-N$ sub-optimal arms in each exploration cycle and adaptively switches between exploration and exploitation. Such design avoids unnecessary exploration of sub-optimal arms and successfully eliminates the dependence on $K$ in the regret, presenting a more suitable EE trade-off strategy in the matching market scenario.
>
>
> - Bandit in matching markets
>
> The application of bandit algorithms in matching markets addresses practical scenarios where learning and optimization under uncertainty are critical. For example, many online platforms, such as job portals, dating apps, and online marketplaces, require iterative matching between users (e.g., job seekers and employers, buyers and sellers). These platforms often operate in environments with incomplete information and evolving preferences. Bandit algorithms can help these platforms learn and adapt to user preferences over time, improving the efficiency and quality of matches.
>
> We acknowledge that our current setting may not perfectly cover real applications such as those involving asynchronous agents, contextual information, and switching costs. However, as a foundational framework, it represents a crucial aspect of learning through interaction with both players and arms. The corresponding algorithm in this basic setting also provides solutions for more general markets.
>
>
>
> - Extension to decentralized markets
>
> Our Algorithm 1 (AOGS) is designed for the decentralized setting and Algorithm 2 is for the centralized setting. For reader-friendly, the algorithm chart of AOGS only summarizes its centralized version. To extend to the decentralized setting, we can modify Algorithm 1 as a phased version like Kong and Li (2023) [17]. Specifically, the total horizon can be separated into several phases with the length of each growing exponentially.
> At the beginning of each phase, players would decide whether to eliminate sub-optimal arms (Line 6-8), and whether to determine one arm as the most preferred one (Line 9-11, respectively). Each player then communicates with others about their current exploration status based on the available observation and then updates their deletion set, available arm
> set, as well as the exploration status (Line 12-19). All players can have a pre-agreed selection ordering for the next phase's exploration (based on the first paragraph of Section 4.1). Within the next phase, players only need to explore arms based on this pre-agreed ordering. In this way, the AOGS algorithm can be generalized to the decentralized setting without influencing the regret order. More details can also be found in the first two paragraphs of Section 4.1.

---

> > ### Comment · Reviewer_SWbh · 2024-08-09
> >
> > Thanks! I've read your response and I've decided to maintain my scores.

---

### Official Review · Reviewer_unKj · 2024-07-14

**Soundness:** 3
**Presentation:** 3
**Contribution:** 2
**Rating:** 5
**Confidence:** 2

**Summary:**

This work studies the problem of bandit learning in two-sided matching markets, where the number of players $N$ is smaller than the number of arms $K$. The players' preferences $\mu\_{i,j}$ are unknown but the arms' utility preferences $\pi\_{i,j}$ are known. Two algorithms with improved regret bounds are proposed. The first one is adaptive online Gale-Shapley (AOGS) for general markets, where multiple stable matchings exist. In AOGS, each player follows an adaptive explore-then-commit strategy to identify the best available arm based on the confidence bounds of each arm reward estimates and whether an arm is better matched to other players. The regret bound of AOGS is stated in terms of a new notion of gap. The second algorithm is centralized UCB combined with offline Gale-Shapely for markets with $\alpha$-condition, where a unique stable matching exists.

**Strengths:**

- Novelty: The following contributions of the work seem novel: the new definition of the gap, the exploration strategy in Algorithm 1, and the improved analysis of Algorithm 2.
- Significance: All algorithms appear to be rigorously analyzed. No proofs are missing. The bound in Theorem 6.2 have optimal dependency on $K, N$ and $log(T)$ compared to a known lower bound.
- Writing quality: The paper is generally well-written.

**Weaknesses:**

The contributions are rather incremental. Details are below:
- The new bound in Theorem 4.1 depends on a new notion of gap. Given that the existing literature in Table 1 already have at least three different notions of gaps, it is not clear in what sense another type of gap is better than the existing ones. Does the new gap capture the difficulty of practical scenarios better? This was not sufficiently discussed in the paper.
- A lower bound result for the new gap is missing. As the paper acknowledged, comparing with the lower bound in Sankararaman et al is not entirely meaningful because the lower bound there uses a different kind of gap.
- In general, a comprehensive comparison with existing results is missing. For example, do any of the existing results in Table 1 imply anything about the new results in the current paper? Another thing is Algorithm 2 was proposed by Liu et al for $\mathrm{gap}\_2$, while the paper uses the same algorithm for $\mathrm{gap}\_3$ under $\alpha$-condition, so it is not clear why the analysis in this paper is considered an improved version of the one in Liu et al (the goal is different). Maybe if the proposed analysis implies that Algorithm 2 obtain optimal bounds on multiple types of gaps *simultaneously*, then it would be a significant result.

**Questions:**

Please see the questions in the Weakness section above. One more question below:
- The assumption that $\pi\_{i,j}$ are known seem strong and has been exploited thoroughly in previous works. How are the results impacted if this assumption does not  hold?

**Limitations:**

- The paper relies on a strong assumption that $\pi\_{i,j}$ must be fully known and does not discuss how the approach and results change if this assumption is violated. In particular, it is not clear whether a weaker assumption would work, such as only the top $k$ $\pi\_{i,j}$ are known for $k < \min(N, K)$.
- The theory in this paper concerns a practical setting that might have some potential societal impacts, none of which is significant.

---

> ### Author Rebuttal · Authors · 2024-08-07
>
> We thank the reviewer unKj for the valuable comments. Please find our response below.
>
> - New gap definition in Theorem 4.1, lower bound for the gap
>
> Recall that the regret dependence on the gap is $1/\Delta^2$, which means the larger the gap, the better the regret. Though we define a new gap in Theorem 4.1, this gap is larger than all of the existing works for the general one-to-one markets (only the result of [19] depends on a larger gap than ours but they have a strong assumption of known $\Delta$), implying that our result not only exhibits better dependence on market sizes but also improves dependence on \( \Delta \).
>
> When players know their exact preferences, they can follow the offline Gale-Shapley algorithm, selecting arms based on its preference ranking one by one until reaching the player-optimal stable matching. So in this sense, each player $p_i$ only uses the ranking of the top $\sigma_i(m_i^*)$. Correspondingly, our new gap definition $\mathrm{gap}_4$ characterizes the difficulty of learning this ranking information in the online setting. Regarding the lower bound, in the problem instance used to derive the lower bound [27], the stable arm of each player is simply their most preferred arm. In this case, our defined $\mathrm{gap}_4$ in Theorem 4.1 is equivalent to the original $\mathrm{gap}_1$. It is unclear whether the lower bound depends on either of these gaps in general markets. We leave this as an interesting future direction.
>
>
>
> - Comparison with existing works
>
> While the lower bound under different gaps remains unclear, we emphasize that our derived regret bounds strictly improve existing works regarding both market sizes and the gap. Specifically, for general markets, the $\mathrm{gap}_4$ in Theorem 4.1 is larger than that in all existing works (except for [19] which assumes known $\Delta$), and the dependence on market size also improves existing works in common cases where $N$ is much smaller than $K$. When markets satisfy $\alpha$-condition, our $\mathrm{gap}_3$ in Theorem 6.2 is larger than the original $\mathrm{gap}_2$ for the centralized UCB algorithm [19], simultaneously removing the dependence on $K$.
>
>
>
>
>
> - Assumption of known arms' preferences
>
> We first want to clarify that ``known arms preferences" means that the arms know their preference rankings, but the players still do not have this information.
> We consider the scenario with one-sided unknown preferences and aim to improve the learning efficiency towards the player-optimal stable matching. In existing works of this line, arms' preference rankings are used to determine whether an arm is a player's possible stable arm (by letting multiple players select the arm and observe who is accepted). If the arms' preferences are also uncertain, players cannot directly ascertain this information in a single round, and the convergence results of existing works do not hold.
> We agree that exploring the two-sided uncertainty is an important direction and some studies have begun to delve into these domains (listed below). Nevertheless, for the currently considered fundamental framework, the optimality problem remains open. We hope to first develop a deep understanding of this fundamental setting and perform a tighter analysis before expanding our focus to encompass generalizations of the setting.
>
>
> Pagare, Tejas, and Avishek Ghosh. "Two-Sided Bandit Learning in Fully-Decentralized Matching Markets." ICML 2023 Workshop The Many Facets of Preference-Based Learning. 2023.
>
> Pokharel, Gaurab, and Sanmay Das. "Converging to Stability in Two-Sided Bandits: The Case of Unknown Preferences on Both Sides of a Matching Market." arXiv preprint arXiv:2302.06176 (2023).

---

> > ### Comment · Reviewer_unKj · 2024-08-08
> >
> > Thanks for the clarification, especially on the double-sided uncertainty part. However, while the paper and the rebuttal repeatedly claim that proving a lower bound should require a separate paper, no clear explanation has been given so far. Can the authors clarify exactly what are the difficulties in adapting the lower bound construction in Sankararaman et al. to your new gap? I am not asking for a formal proof, just high-level ideas on why a lower bound for your gap would require a fundamentally new construction, and what a difficult instance looks like in your setting.

---

> > > ### Author Response · Authors · 2024-08-10
> > >
> > > We thank the reviewer for taking the time to read our response.
> > >
> > > Regarding the lower bound, we want to discuss the instance construction intuition from the view of the total number of rounds required to reach the player-optimal stable matching. In the instance of Sankararaman et al. [27], the player-optimal stable arm of each player is exactly their first ranked arm. So it cannot distinguish between their $\mathrm{gap}_1$ and our $\mathrm{gap}_4$. To derive a lower bound depending on $\mathrm{gap}_4$, we hope to find an example such that $\sigma_i(m_i^*)\neq 1$ and any change in the rankings within the first $\sigma_i(m_i^*)+1$ positions would result in a different player-optimal stable matching.
> > > For example, in the following market, the player optimal stable matching is $\{ (p_1,a_4), (p_2,a_1), (p_3,a_2), (p_4,a_3) \}$. However, if we adjust the $p_1$'s preference ranking as $a_1>a_2>a_4>a_3>a_5$, the player-optimal stable matching under the changed preferences is $\{(p_1, a_4), (p_2, a_3), (p_3, a_1), (p_4,a_2)\}$, which is not stable in the original environment as $p_1$ and $a_3$ form a blocking pair. Although $\mathrm{gap}_1$ (the difference between $a_4$ and $a_5$) of $p_1$ may have already been identified in this scenario, it is insufficient to achieve the optimal stable matching. That's to say, the player $p_i$ must identify the first $\sigma_i(m_i^*)+1$ gaps to find the true optimal stable matching. And the hardness would depend on our newly defined $\mathrm{gap}_4$. To identify these gaps, each player $p_i$ must select the first $\sigma_i(m_i^*)+1$ arms for sufficient times. We hope to find an instance in which these arms prefer other players than this player, such as arm $a_1$ preferring others over $p_1$, forcing $p_1$ to wait for $\Omega(N\log T/\Delta^2)$ rounds for the opportunity to select $a_1$.
> > >
> > >
> > >
> > > $p_1: a_1>a_2>a_3>a_4>a_5,$
> > >
> > > $p_2: a_2>a_3>a_1>a_4>a_5,$
> > >
> > > $p_3: a_3>a_1>a_2>a_4>a_5,$
> > >
> > > $p_4: a_1>a_2>a_3>a_4>a_5;$
> > >
> > > $a_1: p_2>p_3>p_4>p_1,$
> > >
> > > $a_2: p_3>p_4>p_1>p_2,$
> > >
> > > $a_3: p_4>p_1>p_2>p_3,$
> > >
> > > $a_4: p_1>p_2>p_3>p_4,$
> > >
> > > $a_5: p_1>p_2>p_3>p_4.$

---

> > > > ### Comment · Reviewer_unKj · 2024-08-10
> > > >
> > > > This is good! I will raise my score. I encourage the authors to add this lower bound response to the paper.

---

> > > > > ### Author Response · Authors · 2024-08-11
> > > > >
> > > > > We thank the reviewer’s consideration in raising the score. We will include the discussion in the next version of the paper.

---

### Decision · Program_Chairs · 2024-09-25

**Decision:**

Accept (poster)

**Comment:**

The paper proposed a new algorithm for bandit learning in matching markets with regret analysis. It offers new insights into balancing exploration and exploitation in matching markets. The numerical experiments demonstrate significant improvements over existing policies. Please include the discussion on the lower bound and emphasize the precise list of contributions in the final version.